# Counterfactual Vision-and-Language Navigation: Unravelling the Unseen

**Amin Parvaneh, Ehsan Abbasnejad, Damien Teney,**
**Javen Qinfeng Shi, Anton van den Hengel**

Australian Institute for Machine Learning
University of Adelaide, Australia
{amin.parvaneh, ehsan.abbasnejad, damien.teney,
javen.shi, anton.vandenhengel}@adelaide.edu.au

## Abstract

The task of vision-and-language navigation (VLN) requires an agent to follow text instructions to find its way through simulated household environments. A prominent challenge is to train an agent capable of generalising to new environments at test time, rather than one that simply memorises trajectories and visual details observed during training. We propose a new learning strategy that learns both from observations and generated *counterfactual* environments. We describe an effective algorithm to generate counterfactual observations on the fly for VLN, as linear combinations of existing environments. Simultaneously, we encourage the agent's actions to remain stable between original and counterfactual environments through our novel training objective – effectively removing spurious features that would otherwise bias the agent. Our experiments show that this technique provides significant improvements in generalisation on benchmarks for Room-to-Room navigation and Embodied Question Answering.

## 1 Introduction

Deep learning has generated significant advances in computer vision and natural language processing. The most striking successes are witnessed on perceptual tasks that essentially amount to pattern matching. A strength of deep learning is its ability to pick up statistical patterns in large labeled datasets. As a flip side, this capacity leads to models that indiscriminately rely on dataset biases and spurious correlations as much as task-relevant features. This limits the generalisation capabilities of learned models and restrict their applicability on complex tasks (e.g. [1, 2] with images and [3, 4, 5, 6] in multimodal tasks). Most successful applications of deep learning rely on settings where the *seen* training data and the *unseen* test data are statistically similar. Yet we argue that better generalisation could be achieved with new training strategies. This is particularly relevant to multimodal, high-level tasks where training examples can only cover a tiny part of the input space.

In this paper, we propose to *consider the unseen* to learn representations that lead to better generalisation. The method is applied to the task of vision-and-language navigation (VLN, [7, 8, 9]) which requires relating complex inputs with observations of unseen environments. In VLN, an agent receives instructions in natural language and it must decide on a sequence of actions (e.g. *turn left*, *move forward*, ...) to reach a target location while observing 2D images of its environment. The task is extremely ambitious: the agent must learn to ground language with visual observations, to understand sequences of instructions and high-level actions (e.g. *wait by the door*), to generate navigation plans, etc. The standard approach is to train an agent with a combination of reinforcement learning [10, 11] and imitation learning with human-generated examples of instructions and trajectories. These agents can memorise successful sequences of actions and grounding associations but they often fail to apply their capabilities to unseen environments at test time [11]. Our intuition is that a mechanism to reason about alternative observations and trajectories during training could help learning robust navigation strategies. We would like to consider, for example, *what would happen if a desk were observed instead of a chair ?*

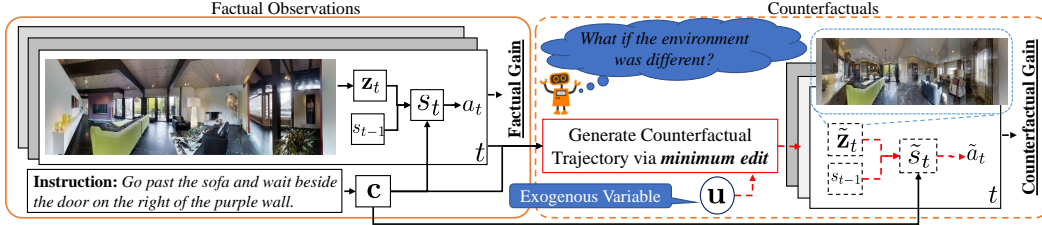

Figure 1: We seek to improve a VLN agent's capability to generalise to unseen environments at test time. Agents are typically trained by reinforcement and imitation learning, using ground-truth pairs of instructions/trajectories ("factual observations", left). We propose to generate alternative, *counterfactual* training observations with combinations of existing environments. We determine the minimum intervention on the factual data that causes the current model to produce different outputs. We then formulate our novel training objective to best exploit these additional examples and improve its generalisation capabilities (right). The generation process is formalised with a causal model of the data, in which we introduce the interpolation coefficients as an exogenous variable $\mathbf{u}$, effectively modelling an intervention on the environment.

Various methods have been proposed to improve generalisation in VLN, such as feature and environment dropout [11], fine-tuning based on the exploration of unseen environments [10, 12] or using beam search [12, 13]. The method we propose is inspired by the framework of counterfactual reasoning [14]. Counterfactuals serve to reason about unobserved scenarios and to estimate the effect of an intervention not represented in the data. In the context of VLN, we essentially want to consider during training *what if we observed a different environment*. Throughout this paper, we call *counterfactuals* training environment examples that we could have observed. We consider the causal model underlying the training environments and introduce an *exogenous* variable that governs their visual features yet is unobserved. We utilise this variable in generating counterfactuals. Intuitively, this exogenous variable captures variations in visual features in the environments that are rather insignificant for the decision making of the agent and can be ignored. At each training iteration, we generate counterfactuals that represent the minimum edit of an existing training data that causes the model to change its action. Thereafter, we formulate a novel objective that encourages the agent to learn from both observed training data and their counterfactuals by explicitly removing the effects of intervention in the agent's policy (see Fig. 1). By introducing additional variations in the observations during training, we encourages the model to rely less on idiosyncrasies of a given environment, and rather learn a policy that better generalises to unseen environments at test time.

The contributions of this paper are summarized as follows.

- We propose a novel training strategy for VLN that generates counterfactuals on the fly to account for unseen scenarios. Using both training data and their counterfactuals, we improve agent's capabilities to generalise to new environments at test time.

- We formalise the new procedure with a causal generative view of the data, in which we introduce an exogenous variable representing interpolation coefficients between original training examples. We derive an efficient algorithm to generate counterfactual instances that represent minimum interventions over original examples that cause the model to change its output.

- We implement the technique on top of a VLN agent for both reinforcement and imitation learning. Experiments on benchmarks for Room-to-Room (R2R) navigation [8] and Embodied Question Answering [9] show significant improvements. We reduce the success rate gap between seen and unseen environments in R2R from about 8% to less than 2.5%.

## 2  Related Work

**Vision and Language Navigation (VLN)** has gained popularity in various forms (instruction following [8, 15], object or room probing [16, 17], embodied question answering [9, 18], vision and language dialogue [7, 19]). Generalisation to unseen environments remains an unsolved challenge, despite techniques like enhanced features and beam search, panorama view [12], attention mechanisms [13], and other heuristics [10, 20, 21]. Environment Dropout [11] randomly drops visual features to simulate variations in environments. Our approach does not require access to held-out trajectories, which may not be available in other tasks (rather than R2R). Our method can be used in a variety of tasks, as demonstrated with EQA in the experiments.

Principles of **counterfactual reasoning** [14, 22] have been applied beyond standard causal inference to augment training in bandit settings [23], and in recommendation [24] and explanation systems [25]. Kaushik et al. [26] proposed a human-in-loop process to augment datasets with counterfactual instances. In reinforcement learning [27, 28], counterfactuals are used in off-policy settings to improve sample efficiency. Our technique is also related to adversarial training [29, 30, 31, 32] in that we generate variations of training examples that cause the current model to switch its predictions. The major difference is that our approach provides alteration to the input, or rather its representations, by a variable that is conditioned on the real training data rather than a simple perturbation.

Using counterfactuals for VLN was explored in [33] in which adversarial paths that are hard for the policy to navigate are generated. Our approach differs from their adversarial augmentation method in that intervene in visual features rather than focusing on difficult trajectories. Our method, while being simpler, outperforms theirs with almost 10% in success rate.

The closest work to this one is [34]. The authors generate counterfactual data using interpolations for vision-and-language tasks, including visual question answering. The differences with this work are that (1) we only intervene on visual features, (2) we backpropagate the loss in counterfactual environments instead of using it as a change ratio for factual loss calculation, and (3) we explicitly focus on removing the effects of intervention. Our work also extensively focuses on VLN.

In comparison to standard **data augmentation**, our counterfactual instances do not rely on hand-crafted or domain-specific rules, and they are generated on the fly. MixUp [35, 36] performs data augmentation with interpolations and label smoothing. Mixup is not directly applicable to VLN since (1) VLN is sequential in nature, (2) an interpolation of state-action from one trajectory to another may lead to catastrophic difference in the objective. Our approach intervenes in the visual features to simulate the agent's behaviour in a counterfactual environment, where the agent still has to follow the same instruction and sequence of actions

## 3 Methodology

### 3.1 Problem Definition

Our task is to train an agent capable of grounding a command, in the form of natural language, to the current visual view and taking suitable actions that lead to the target location. Formally, the agent is given natural language instructions or commands as a sequence of words $\mathbf{c} = [w_1, w_2, .., w_L]$ to be executed in the environment $\mathcal{E}$. We consider all the instructions to be in a set $\mathcal{C}$. The process can be viewed as a Partially Observable Markov Decision Process (POMDP) where a trajectory is a sequence of length $T$ of observation $\mathbf{o}_t$, state $\mathbf{s}_t$ and action $a_t$ for each time step $t$ i.e. $\tau = \{\mathbf{o}_1, \mathbf{s}_1, a_1, \ldots, \mathbf{o}_T, \mathbf{s}_T, a_T\}$. The probability of each trajectory given the instruction is[1]

$$\pi_{\boldsymbol{\theta}}(\tau \,|\, \mathbf{c}) \;=\; \prod_{t=1}^{T} p(a_t \,|\, \mathbf{s}_t)\, p(\mathbf{s}_t \,|\, \mathbf{s}_{t-1}, \mathbf{z}_t, \mathbf{c})\, p(\mathbf{z}_t \,|\, \mathbf{o}_t)\,. \tag{1}$$

Here, $\pi_{\boldsymbol{\theta}}$ is the agent's *policy* (Unless explicitly mentioned otherwise, $\boldsymbol{\theta}$ represents all parameters which is omitted from the right-hand side probabilities for brevity). In the visual navigation scenario we consider, $\mathbf{o}_t$ as the visual observation of the scene in which the agent is, $\mathbf{s}_t$ as a representation of the trajectory history[2] and $a_t$ as the chosen action at time $t$ (e.g. `turn left` or `stop` for when the trajectory is finished). By convention, $\mathbf{s}_0$ is a sample from the state prior (e.g. uniform). We denote a latent representation of the visual scene by $\mathbf{z}$ and assume it is obtained using a function $\mathbf{z} = f_{\mathbf{o}}(\mathbf{o})$, e.g. a pretrained CNN for the visual inputs, thus $p(\mathbf{z}_t \,|\, \mathbf{o}_t) = \delta(\mathbf{z} - f_{\mathbf{o}}(\mathbf{o}))$ where $\delta$ is the Dirac delta.

**Training with imitation learning and reinforcement learning.** The common practice in visual navigation is to use a training set $\mathcal{D} = \{(\tau_i, \mathbf{c}_i)\}_{i=1}^{n}$ containing human-provided trajectories and instructions. This training set is used in supervised learning to bootstrap the agent' behaviour through cloning human's actions. In addition, reinforcement learning is used so that the agent learns from the environment's feedback. The training procedure optimises the following objective [11]:

$$\max_{\boldsymbol{\theta}} \quad \underbrace{\mathbb{E}_{(\tau, \mathbf{c}) \sim \mathcal{D}} \big[ \log \pi_{\boldsymbol{\theta}}(\tau \,|\, \mathbf{c}) \big]}_{\mathcal{G}_{\text{IL}}(\boldsymbol{\theta})} \;+\; \lambda \, \underbrace{\mathbb{E}_{\mathbf{c} \sim \mathcal{C}} \big[ \, \mathbb{E}_{\tau \sim \pi_{\boldsymbol{\theta}}(\tau \,|\, \mathbf{c})}[R(\tau)] \, \big]}_{\mathcal{G}_{\text{RL}}(\boldsymbol{\theta})} \,. \tag{2}$$

The first term $\mathcal{G}_{\text{IL}}(\boldsymbol{\theta})$ is a simple log-likelihood of human-provided examples using Eq. (1) (imitation learning). The second term $\mathcal{G}_{\text{RL}}(\boldsymbol{\theta})$ corresponds to the execution of the policy in the environment

and receiving a reward $R(\tau)$. The hyperparameter $\lambda$ serves to balance the importance of imitation learning versus reinforcement learning. The reward captures the agent's success in navigating the environment. In a Room-to-Room navigation task, the reward is a combination of a large positive number for reaching the target location at the end of each episode, and a small positive/negative number for reducing/increasing the distance to that location at each step. To update the parameters of the policy during RL, we employ an on-policy algorithm such as actor-critic [37].

## 3.2 Counterfactual Formulation in VLN

The state variable $\mathbf{s}$ ideally is the representation of the history of observations and actions. The final decision of the agent is taken conditioned on this variable and as such is of great importance. However, as is common with other multi-modal problems (e.g. VQA [6, 4]) this variable captures particular biases and regularities in the input and may even ignore important patterns which significantly limits the generalisation ability of the agent. To remedy the situation, we consider an exogenous variable that intervenes the observations. By introducing and reasoning about this variable, the agent is encouraged to consider alternative observations and representations. In addition, the agent obtains the capacity to reason about "what if" the observations were different.

To that end, we consider the counterfactual distribution of the trajectory where each observation is replaced by its intervened alternative $\tilde{\mathbf{z}}_t^{\mathbf{u}}$:

$$\tilde{\pi}_{\boldsymbol{\theta}}(\tilde{\tau} \,|\, \mathbf{c},\, \mathbf{u}) \;=\; \prod_{t=1}^{T} p(a_t \,|\, \tilde{\mathbf{s}}_t)\, p(\tilde{\mathbf{s}}_t \,|\, \tilde{\mathbf{s}}_{t-1},\, \tilde{\mathbf{z}}_t^{\mathbf{u}},\, \mathbf{c}). \tag{3}$$

In this distribution, the conditional dependence on the scene observations $\mathbf{o}_t$ is suppressed because of the intervention. We denote with $\tilde{\tau}$ the trajectories obtained by replacing a given embedding of the visual scene $\mathbf{z}_t$ with its counterfactual $\tilde{\mathbf{z}}_t^{\mathbf{u}}$ based on the influence of $\mathbf{u}$. Imagine that the agent observes a chair that represents an obstacle to be avoided. A counterfactual situation would ask, for example "what if the agent observed a table?". The exogenous variable is conditioned on the factual trajectories observed in the training set. The expectation with respect to the exogenous variable serves to consider a whole range of possible alternatives. The expected reward for counterfactual trajectories $\tilde{\mathcal{G}}_{\mathrm{RL}}(\boldsymbol{\theta})$ (to be compared with $\mathcal{G}_{\mathrm{RL}}(\boldsymbol{\theta})$ of Eq. (2)), is obtained from the states intervened based on the exogenous variable $\mathbf{u}$:

$$\tilde{\mathcal{G}}_{\mathrm{RL}}(\boldsymbol{\theta}) := \mathbb{E}_{(\tau, \mathbf{c}) \sim \mathcal{D}}\Big[\, \mathbb{E}_{\mathbf{u} \sim p(\mathbf{u} \,|\, \tau, \mathbf{c})}\big[\, \mathbb{E}_{\tilde{\tau} \sim \tilde{\pi}_{\boldsymbol{\theta}}(\tilde{\tau} \,|\, \mathbf{c},\, \mathbf{u})}[R(\tilde{\tau})]\,\big]\,\Big] \tag{4}$$

$$\tilde{\mathcal{G}}_{\mathrm{IL}}(\boldsymbol{\theta}) := \mathbb{E}_{(\tau,\, \mathbf{c}) \sim \mathcal{D}}\Big[\, \mathbb{E}_{\mathbf{u} \sim p(\mathbf{u} \,|\, \tau, \mathbf{c})}\big[\, \log \tilde{\pi}_{\boldsymbol{\theta}}(\tilde{\tau} \,|\, \mathbf{c},\, \mathbf{u})\,\big]\,\Big]$$

We detail $p(\mathbf{u} \,|\, \tau, \mathbf{c})$ and how to generate counterfactuals using $\tilde{\pi}_{\boldsymbol{\theta}}(\tilde{\tau} \,|\, \mathbf{c},\, \mathbf{u})$ in Section 3.3.

The differences between $\mathcal{G}_{\mathrm{RL}}(\boldsymbol{\theta})$ and $\tilde{\mathcal{G}}_{\mathrm{RL}}(\boldsymbol{\theta})$ as well as between $\mathcal{G}_{\mathrm{IL}}(\boldsymbol{\theta})$ and $\tilde{\mathcal{G}}_{\mathrm{IL}}(\boldsymbol{\theta})$ correspond to the Conditional Average Treatment Effect (CATE) [23]. These differences reflect how the intervention influences the reward and log-likelihood. They are defined as

$$\Delta_d = \mathcal{G}_{\mathrm{IL}}(\boldsymbol{\theta}) - \tilde{\mathcal{G}}_{\mathrm{IL}}(\boldsymbol{\theta}) \qquad \text{and} \qquad \Delta_\tau = \mathcal{G}_{\mathrm{RL}}(\boldsymbol{\theta}) - \tilde{\mathcal{G}}_{\mathrm{RL}}(\boldsymbol{\theta}) \,. \tag{5}$$

We want to optimise our agent such that, after learning from the training set, performs similarly when faced with unobserved alternative scenarios. In other words, we want $\Delta_\tau$ and $\Delta_d$ to be small. This effectively reduces the influence of interventions and as such discourages bias to spurious features. We add, to the objective of Eq. (2), constraints on the magnitude of $\Delta_d$ and $\Delta_\tau$:

$$\max_{\boldsymbol{\theta}} \quad \mathcal{G}_{\mathrm{IL}}(\boldsymbol{\theta}) + \lambda\, \mathcal{G}_{\mathrm{RL}}(\boldsymbol{\theta}) \quad \text{s.t.} \quad \Delta_\tau \leq \epsilon_\tau \quad \text{and} \quad \Delta_d \leq \epsilon_d \,, \tag{6}$$

with $\epsilon_d$ and $\epsilon_\tau$ small constants. Introducing the Lagrange multipliers $\alpha$ and $\beta$, we have

$$\max_{\boldsymbol{\theta}} \quad (1 - \alpha)\, \mathcal{G}_{\mathrm{IL}}(\boldsymbol{\theta}) \,+\, \alpha\, \tilde{\mathcal{G}}_{\mathrm{IL}}(\boldsymbol{\theta}) \,+\, (\lambda - \beta)\, \mathcal{G}_{\mathrm{RL}}(\boldsymbol{\theta}) \,+\, \beta\, \tilde{\mathcal{G}}_{\mathrm{RL}}(\boldsymbol{\theta}) \,. \tag{7}$$

We assume $\beta = \alpha\lambda$ and $(1 - \alpha) > 0$ for simplicity, which gives the final objective:

$$\max_{\boldsymbol{\theta}} \quad \underbrace{\big(\mathcal{G}_{\mathrm{IL}}(\boldsymbol{\theta}) + \lambda\, \mathcal{G}_{\mathrm{RL}}(\boldsymbol{\theta})\big)}_{\text{Original navigation}} \;+\; \tfrac{\alpha}{(1-\alpha)}\, \underbrace{\big(\tilde{\mathcal{G}}_{\mathrm{IL}}(\boldsymbol{\theta}) + \lambda\, \tilde{\mathcal{G}}_{\mathrm{RL}}(\boldsymbol{\theta})\big)}_{\text{Counterfactual navigation}} \,. \tag{8}$$

Technically, when increasing $\alpha/(1 - \alpha)$, we choose to give more weight to what could have been seen (variations in the environment) rather than maximising the gain. Therefore, when the trajectories are longer we need smaller $\alpha/(1 - \alpha)$ which intuitively allows the model to focus on correct actions at each state rather than variations that could have been observed. Note, learning longer trajectories are generally harder and a small mistake has more significant impact. This novel objective is used with the counterfactuals, of which we next discuss the generation.

### 3.3 Counterfactual Distribution Learning and Generation

Computing Eq. (4) hinders on: (1) the distribution of the counterfactual trajectories given the intervention by exogenous variable $\tilde{\pi}_{\boldsymbol{\theta}}(\tau|\mathbf{u}, \mathbf{c})$, (2) the conditional of the exogenous $p(\mathbf{u}|\tau, \mathbf{c})$ given the observed trajectory-instruction pair from data, and (3) combining (1) and (2) to have the probability of the counterfactual trajectory as $\tilde{\pi}_{\boldsymbol{\theta}}(\tau \mid \mathbf{c}) = \mathbb{E}_{p(\mathbf{u}\mid\tau, \mathbf{c})}[\tilde{\pi}_{\boldsymbol{\theta}}(\tau \mid \mathbf{c}, \mathbf{u})]$. Here, $\mathbf{u}$ is marginalised out to remove the impact of the intervention or spurious features.

1. **Sampling from $\tilde{\pi}_{\boldsymbol{\theta}}(\tau|\mathbf{c}, \mathbf{u})$:** To sample a counterfactual trajectory, we first sample a pair of real trajectories from the observations such that at least one has the language instruction, i.e. $\{(\tau, \mathbf{c}), (\tau', \mathbf{c}')\} \sim \mathcal{D}$. Subsequently, we choose the counterfactual visual features to be a linear interpolation. Given a sample $\mathbf{u} \in [0, 1]^d$ ($d$ being the dimensionality of $\mathbf{z}$) with slight abuse of notation, we have:

$$\tilde{\tau} = \{\tilde{\mathbf{z}}_0^{\mathbf{u}}, \tilde{\mathbf{s}}_0, a_0, \ldots, \tilde{\mathbf{z}}_T^{\mathbf{u}}, \tilde{\mathbf{s}}_T, a_T\} \sim \tilde{\pi}_{\boldsymbol{\theta}}(\tau|\mathbf{u}, \mathbf{c}), \quad \tilde{\mathbf{z}}_t^{\mathbf{u}} = \mathbf{u} \odot \mathbf{z}_t + (\mathbf{1} - \mathbf{u}) \odot \mathbf{z}_t', \quad (9)$$
$$\text{with} \quad \mathbf{z}_t = f_{\mathbf{o}}(\mathbf{o}_t), \ \mathbf{z}_t' = f_{\mathbf{o}}(\mathbf{o}_t'), \quad \mathbf{o}_t \in \tau, \ \mathbf{o}_t' \in \tau'.$$

We use $\odot$ to represent an element-wise product. When the length of the second trajectory $\tau'$ is shorter, we choose to repeat its final visual features for interpolation. Alternative approaches such as generative adversarial networks [38] could be employed, albeit our simple option presents a clear advantage in computational efficiency.

2. **Exogenous variable's distribution $p(\mathbf{u} \mid \tau, \mathbf{c})$:** Given the prior $p(\mathbf{u})$, we have $p(\mathbf{u} \mid \tau, \mathbf{c}) \propto p(\mathbf{u})\tilde{\pi}_{\boldsymbol{\theta}}(\tau \mid \mathbf{c}, \mathbf{u})$ as the posterior. It is easy to see that with our definition in Eq. (9), when $\mathbf{u} = 1$ we uncover $\pi_{\boldsymbol{\theta}}(\tau \mid \mathbf{c})$ in Eq. (1). In other words, $\mathbf{u} = 1$ provides the max-likelihood since that gives rise to an observed trajectory. We consider a Beta distribution for the prior.

3. **Finding minimum interventions that change the agent's decision:** Having (1) and (2) we can sample a counterfactual trajectory $\tilde{\pi}_{\boldsymbol{\theta}}(\tau \mid \mathbf{c})$ (with $\mathbf{u}$ marginalised out). One can resort to MCMC or a variational lower bound to sample the most likely counterfactual. However, in the interest of efficiency and simplicity, we choose the exogenous variable with the highest likelihood that produces the most likely counterfactual. In other words, we seek the minimum intervention (i.e. minimum edit) that changes the agent's decision (remember, we want our counterfactuals to be very different from observations). Since changing the agent's decision may lead to a different route in the environment, we additionally constrain the counterfactual trajectory to have the same instructions. Given a training example $(\mathbf{c}, \tau)$, the following optimisation identifies such an intervention parametrised by $\mathbf{u}$ (note $\tilde{\tau}$ is the counterfactual of $\tau$):

$$\max_{\mathbf{u} \in [0,1]^d} \quad p(\mathbf{u} \mid \tau, \mathbf{c}) + \log p(\mathbf{c} \mid \tilde{\tau}, \boldsymbol{\phi}) \tag{10}$$
$$\text{s.t.} \quad a_t' \neq a_t \ \forall \ t \quad \text{with} \quad a_t' = \arg\max_{a_t} p(a_t \mid \tilde{\mathbf{s}}_t) \, p(\tilde{\mathbf{s}}_t \mid \tilde{\mathbf{s}}_{t-1}, \tilde{\mathbf{z}}_t^{\mathbf{u}}, \mathbf{c}).$$

The second term in Eq. (10) measures how likely an instruction is for a trajectory for which we utilise the speaker model of [12] with parameters $\boldsymbol{\phi}$. The optimisation of Eq. (10) is too expensive to perform for every training trajectory. We note that the first term is maximised when $\mathbf{u}$ is close to one, as such a relaxed version by turning the constraint into an extra term in the objective is devised:

$$\max_{\mathbf{u} \in [0,1]^d} \quad \|\mathbf{u}\| + \log p(\mathbf{c} \mid \tilde{\tau}, \boldsymbol{\phi}) - \gamma \sum_{t=1}^{T} \Big( \log p(a_t \mid \tilde{\mathbf{s}}_t) + \log p(\tilde{\mathbf{s}}_t \mid \tilde{\mathbf{s}}_{t-1}, \tilde{\mathbf{z}}_t^{\mathbf{u}}, \mathbf{c}) \Big), \tag{11}$$

where $\gamma$ is a hyper-parameter. The first two terms in this equation ensure the intervention is minimal and the counterfactual trajectory is most likely to follow the same instructions. The constraint, on the other hand, finds the counterfactual trajectory by fooling the current policy.

A summary of the whole training algorithm is provided in Algorithm 1.

## 4 Experiments

To show the effectiveness of our counterfactual contemplation approach we applied it to both Room-to-Room (R2R) navigation and Embodied Question Answering (EQA). In all of our experiments, we only intervene in the visual features as discussed in Sec. 3.3. We set the prior $p(\mathbf{u})$ to $\text{Beta}(0.75, 0.75)$, and use 5 interactions to optimise Eq. (11) with the learning rate set to 0.1. Using grid search, we concluded $\gamma = 0.1$ provides best results. We closely follow Algorithm 1 to learn the parameters, more details are provided in the supplement.

**Algorithm 1:** Training of a VLN agent through IL and RL, with factual data (original training set) and counterfactual observations (generated instances).

---

**Inputs:** dataset $\mathcal{D}$, initial policy parameters $\boldsymbol{\theta}^0$, learning rate $\xi_u, \xi_\theta$
**for** $i = 1$ *to max_iterations* **do**
    Pick a sample from the dataset $(\tau, \mathbf{c}) \sim \mathcal{D}$
    Generate exogenous variable from the prior: $\mathbf{u}^0 \sim p(\mathbf{u})$
    Pick another sample from the dataset $(\tau', \mathbf{c}') \sim \mathcal{D}$
    `// use Eq. (11) to get the counterfactual trajectory`
    **for** $j$ *to* $N$ **do**
        $\tilde{\tau} = \{\tilde{\mathbf{z}}_0^{\mathbf{u}}, \tilde{\mathbf{s}}_0, a_0, \ldots, \tilde{\mathbf{z}}_T^{\mathbf{u}}, \tilde{\mathbf{s}}_T, a_T\}$, $\tilde{\mathbf{z}}_t^{\mathbf{u}} = \mathbf{u} \odot \mathbf{z}_t + (\mathbf{1} - \mathbf{u}) \odot \mathbf{z}_t'$ `// Eq. (9)`
        $\mathbf{u}^{j+1} = \mathbf{u}^j + \xi_u \nabla_{\mathbf{u}} \Big( \|\mathbf{u}\| + \log p(\mathbf{c}|\tilde{\tau}, \boldsymbol{\phi}) - \gamma \sum_{t=1}^T \big( \log p(a_t|\tilde{\mathbf{s}}_t) + \log p(\tilde{\mathbf{s}}_t|\tilde{\mathbf{s}}_{t-1}, \tilde{\mathbf{z}}_t^{\mathbf{u}}, \mathbf{c}) \big) \Big)$
    **end**
    $g_{\text{IL}} = \log \pi_{\boldsymbol{\theta}}(\tau \,|\, \mathbf{c}) + \frac{\alpha}{1-\alpha} \log \tilde{\pi}_{\boldsymbol{\theta}}(\tilde{\tau} \,|\, \mathbf{c})$   `// imitation learning gain`

    Given the instruction $\mathbf{c}$, rollout trajectories $\tau_{\text{rl}}$ and $\tilde{\tau}_{\text{rl}}$ from the current navigation policy
     without and with interventions respectively
    $g_{\text{RL}} = \mathbb{E}_{\tau_{\text{rl}} \sim \pi_{\boldsymbol{\theta}}(\tau_{\text{rl}} \,|\, \mathbf{c})}[R(\tau_{\text{rl}})] + \frac{\alpha}{1-\alpha} \mathbb{E}_{\tilde{\tau}_{\text{rl}} \sim \tilde{\pi}_{\boldsymbol{\theta}}(\tilde{\tau}_{\text{rl}} \,|\, \mathbf{c})}[R(\tilde{\tau}_{\text{rl}})]$   `// RL gain`
    $\boldsymbol{\theta}^i = \boldsymbol{\theta}^{i-1} + \xi_\theta \nabla_{\boldsymbol{\theta}} \big( g_{\text{IL}} + \lambda g_{\text{RL}} \big)$   `// update based on Eq. (8)`
**end**

---

## 4.1 Room-to-Room Navigation

**Dataset:** Room-to-Room (R2R) [8] is a dataset of natural language instructions for indoor navigation collected using Amazon Mechanical Turk (AMT) and employing a simulator based on Matterport3D environments [39]. The training is based on $14,025$ pairs of instruction-visual path in 61 environments. The validation is done in two settings: (1) *seen* where the environment is from the training set but the instructions are not and (2) *unseen* where both the instructions and the visual observations are never seen by the agent.

**Implementation details:** We closely follow the experiment setup of [11] where the visual observations consists of the features extracted using the pretrained ResNet-152 [40] from the egocentric panoramic view of the agent. Similarly, the policy is an attention encoder-decoder network that chooses an action from a set of directions at each time-step. Following the approach proposed in [12], our *speaker* is a sequence-to-sequence model which evaluates the likelihood of an instruction for a trajectory. We optimise our models using RMSprop with a learning rate of $1 \times 10^{-4}$ and batch size of 64 for $80,000$ iterations in all of our experiments, except when indicated. Further details are provided in the supplements.

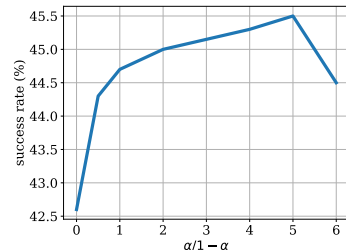

Figure 2: The effect of $\alpha$ on the results inside unseen environments. $\frac{\alpha}{(1-\alpha)} = 0$ means no counterfactual is used (conventional training).

We set $\alpha \approx 0.83$ (i.e. $\frac{\alpha}{(1-\alpha)} = 5$) by grid search in behavioural cloning setting (without counterfactual learning) for all the experiments. Value of $\alpha$ balances the factual and counterfactual and as shown in Fig. 2 increasing it (more weights for counterfactuals) improves the performance in the unseen environments to a point. Increasing it further reduces the generalisation since the agent forgets the factual observations.

**Baselines:** To evaluate our approach, we conduct extensive experiments in different learning settings similar to that of [11, 8] for fair comparison: *imitation learning* (IL; $\lambda = 0$), with additional *reinforcement learning* (IL+RL), and with additional *data augmentation* (IL+RL+Aug). We employ behaviour cloning and advantage actor-critic (A2C) algorithm [37] when IL and RL are needed respectively. The reward is calculated based on the agent's progress toward the target and its final success/failure similar to the baselines (details in the suppl.). In addition, in the augmented setting, similar to [11], we fine-tune our trained model from IL+RL for the maximum of $200,000$ iterations with additional samples obtained from instructions sampled from the speaker.

**Evaluation metrics:** Similar to [8, 11, 20, 12], we employ both the Navigation Error (NE), the difference as measured in meters between the agent's final position and the target location, and the

Success Rate (SR), the the portion of traversed trajectories at which the NE is less than 3 meters, to evaluate the performance of a navigating agent. However, Success weighted by Path Length (SPL) [41] better represents the efficiency by taking into account the inverse ratio of the agent's Trajectory Length (TL)–the distance the agent travelled– to the ground-truth. We demonstrate all of these metrics for both seen and unseen environments.

**Results**: We report the evaluation performance of the proposed approach in Table 1. We indicate the use of counterfactual objective in Eq. (8) by **+Counterfactuals**. We consider reporting the performance of our approach by simply conditioning the interventions on the prior distribution of the exogenous variable indicated by **+Pior**. Using the prior, compared to the one optimised in Eq. (11), evaluates the value of estimating the posterior. As shown, by incorporating the counterfactuals the navigation performance of

| Model | Validation-Seen | | | | Validation-Unseen | | | |
|---|---|---|---|---|---|---|---|---|
| | NL↓ | NE↓ | SR↑ | SPL↑ | NL↓ | NE↓ | SR↑ | SPL↑ |
| Seq-to-Seq [8] | 11.3 | 6.01 | 38.6 | - | 8.4 | 7.81 | 21.8 | - |
| Speaker-Follower [12] | - | 4.86 | 52.1 | - | - | 7.07 | 31.2 | - |
| Co-Grounding [13] | - | 3.65 | 65.0 | 0.56 | - | 6.07 | 42.0 | 0.28 |
| IL* [11] | 9.9 | 5.34 | 50.2 | 0.48 | 9.5 | 6.10 | 42.6 | 0.40 |
| **IL+Prior** | 9.9 | **5.17** | **50.5** | **0.48** | 9.2 | 5.89 | 45.5 | 0.43 |
| **IL+Counterfactuals** | **9.8** | 5.37 | 48.9 | 0.47 | **9.1** | **5.75** | **46.4** | **0.44** |
| IL+RL* [11] | 10.3 | **4.65** | **55.8** | **0.53** | 9.7 | 5.73 | 44.9 | 0.41 |
| **IL+RL+Prior** | 11.2 | 4.78 | 54.0 | 0.51 | 14.9 | 5.52 | 48.5 | 0.44 |
| **IL+RL+Counterfactuals** | 10.7 | 4.75 | 53.6 | 0.51 | 11.8 | **5.42** | **49.4** | **0.46** |
| IL+RL+Aug* [11] | 10.3 | 4.01 | 62.5 | 0.60 | **9.7** | 5.48 | 50.3 | 0.47 |
| **IL+RL+Aug+Prior** | 11.0 | 3.65 | 64.4 | 0.61 | 13.5 | 5.13 | 52.4 | 0.48 |
| **IL+RL+Aug+Counterfactuals** | **10.8** | **3.65** | **68.2** | **0.64** | 12.4 | **4.95** | **53.5** | **0.49** |

Table 1: Evaluation metrics for R2R Navigation. Navigation Length (NL) and Navigation Error (NE) values are represented in meters, while Success Rate (SR) values are percentage.
↑ indicates higher is better, while ↓ shows lower is better.
* Results are as reported in the official implementation: `https://github.com/airsplay/R2R-EnvDrop`.

the imitating agent, in particular for the unseen environments, improves significantly. We particularly observe around $4\%$ improvement in SR and SPL compared to the baseline. More importantly, our method improves the generalisation by decreasing the SR gap between the seen and unseen environments from around $8$ to $2.5\%$–a significant improvement indeed.

Once the reinforcement signal is added (i.e. $\lambda = 5$), our proposed policy's performance improves further by more than $3\%$ for SR compared to its IL counterpart. Furthermore, our method enjoys about $5\%$ improvement in SR and SPL in unseen environments, and, more importantly, an approximately $6.7\%$ drop in the seen versus unseen performance gap. Further, using augmentations, our model enjoys another $4\%$ boost in both SR and SPL.

Finally, we submitted our proposed model to the

| Model | Validation-Unseen | | | | Test-Unseen | | | |
|---|---|---|---|---|---|---|---|---|
| | NL↓ | NE↓ | SR↑ | SPL↑ | NL↓ | NE↓ | SR↑ | SPL↑ |
| Random [8] | 9.8 | 9.23 | 16.3 | - | 9.9 | 9.79 | 13.2 | 0.12 |
| Seq-to-Seq [8] | 8.4 | 7.81 | - | 0.22 | 8.1 | 7.85 | 20.4 | 0.18 |
| Speaker-Follower [12] | - | 6.62 | 35.5 | - | 14.8 | - | 35.0 | 0.28 |
| Self-Monitoring [13] | - | 5.41 | 47.0 | 0.34 | 18.0 | 5.67 | 48.0 | 0.35 |
| Reinforced Cross-Modal [10] | 11.5 | 6.09 | 50.1 | 0.43 | 12.0 | 6.12 | 43.0 | 0.38 |
| Tactical-Rewind [42] | 21.2 | 4.97 | **56.0** | 0.43 | 22.1 | 5.14 | 54.0 | 0.41 |
| Counterfactual VLN [33] | - | 5.40 | 47.7 | 0.43 | - | 5.80 | 45.1 | 0.41 |
| Environment Dropout [11] | 10.7 | 5.22 | 52.2 | 0.48 | 11.7 | 5.23 | 51.5 | 0.47 |
| **Ours** | 12.4 | **4.95** | 53.5 | **0.49** | 13.0 | **4.90** | 54.9 | **0.50** |
| PRESS* [43] | 10.4 | 5.28 | 49.0 | 0.45 | 10.8 | 5.49 | 49.0 | 0.45 |
| PREVALENT* [44] | 10.2 | 4.71 | 58.0 | 0.53 | 10.5 | 5.30 | 54.0 | 0.51 |

Table 2: The comparison of our results with others in unseen environments. Test-unseen results are reported on the task's leaderboard in single-run setting.
* The indicated methods are taking advantage of self-supervised pre-training. Our method can be applied on top of these methods to result in a further improvement.

leaderboard for the evaluation on the test set–a hold-out dataset of 18 environments for a fair challenge[3]. Table 2 demonstrates the superior performance of our model in comparison to other baselines. Interestingly, our model outperforms the EnvDrop model [11], the most similar model to ours, by a significant margin of $3.4$ percent in SR and 3 points in SPL. Besides, our agent surpasses

self-supervised pre-training of [44], in terms of success rate and navigation error–a model that we believe can further benefit from our approach.

## 4.2 Embodied Question Answering

**Dataset**: Embodied Question Answering (EQA) [9] is a challenging variant of Vision and Language Navigation where in contrast to R2R task, the agent is given a general question about an object in the environment, e.g. "what colour is the car?". Spawning in a random location in an unseen environment at test time, the agent must first navigate to the proximity of the desired object and subsequently answer the given question. The dataset consists of $6,912$ tuples of route-question-answer in $645$ distinct training environments and a collection of $898$ tuples in $57$ unseen environments for the test set. At each step, the agent is provided with an egocentric RGB image based on which the agent should choose the next action among a set of 4 discrete choices (`forward`, `turn-left`, `turn-right` and `stop`). We treat the question as the instructions of the R2R dataset.

**Implementation details:** Our navigation policy is a simple 2-layer Gated Recursive Unit (GRU) and visual features are obtained from a 4-layer CNN pre-trained using an auto-encoder from House3D images [9] (details in Supplements). We train all of the models for 30 epochs (more than $10,000$ iterations) in a behavioural cloning setting with a batch size of 20 and learning rate set to $1 \times 10^{-3}$ using Adam optimiser. It should be noted that since there is no instructions to be followed (just the question here) we disregard the second term in Eq. (11) for this task.

**Evaluation metrics:** For the evaluation, we spawn the agent in 10, 30, or 50 steps away from the target location in terms of the shortest path (similar to [9]). The main metric for the evaluation is the distance (in meters) between the location where the agent stops and the ground-truth target denoted by $d_T$. Additionally, we consider $d_\Delta = d_T - d_0$ as another critical metric measuring the overall progress of the agent from its initial position $d_0$ towards the target. In contrast to $d_T$, higher values of $d_\Delta$ show better performance. The agent is constrained to a maximum of 100 steps at each episode.

**Results:** As shown in Table 3, almost $10\%$ increase in generalisation to unseen environments is achieved by letting the agent contemplate the unseen. Finally, not only our approach improves the performance of the agent in reaching short-term goals ($T_{-10}$), but it also enhances its accuracy in finding distant objects ($T_{-50}$).

| Model | $d_T\downarrow$ | | | $d_\Delta\uparrow$ | | |
| --- | --- | --- | --- | --- | --- | --- |
| | $T_{-10}$ | $T_{-30}$ | $T_{-50}$ | $T_{-10}$ | $T_{-30}$ | $T_{-50}$ |
| PACMAN [9] | 1.39 | 4.98 | 9.33 | -0.45 | 0.49 | 1.66 |
| Neural Modular Control [45] | 0.85 | 4.32 | 9.29 | 0.09 | 1.15 | 1.70 |
| GRU | 0.74 | 3.99 | 8.73 | 0.20 | 1.48 | 2.26 |
| **GRU+Prior** | 0.73 | 3.95 | 8.50 | 0.21 | 1.52 | 2.49 |
| **GRU+Counterfactuals** | **0.71** | **3.88** | **8.46** | **0.23** | **1.59** | **2.53** |

Table 3: Evaluation metrics for EQA navigation.

EQA is more complex than R2R (long trajectories and high-level language instructions) for which the scores are generally low and the agent learns trivial actions, e.g. going through the door. We found correspondingly using grid search, the best performance is when $\alpha \approx 0.29$ (i.e. $\frac{\alpha}{(1-\alpha)} = 0.4$)–a considerably smaller value to that of R2R. This supports our hypothesis for using longer trajectories in Eq. (8) in which, when the gain is low, the agent must primarily focus on maximising gain (even if that leads to trivial actions) rather than variations. Nevertheless, using counterfactuals even for such a difficult task improves performance of our agent to achieve state-of-the-art results.

## 5 Conclusions

Generalisation ability is paramount for developing a practical VLN in robots that can operate in the wild, yet many overfit the instructions to the visual stimuli in the training. More importantly, current approaches fail to incorporate any mechanism for reasoning about the likelihood of alternative trajectories – a crucial skill for the task. To remedy the issue, we turned to the counterfactuals as a principled approach for reasoning about unobserved scenarios for estimating the effect of an intervention that is not directly represented in the data. We formulated the new learning objective to incorporate both the real data as well as the counterfactuals obtained conditioned on the exogenous variable. This implicitly forces the navigation policy and the internal state representation to learn semantics and high-level relations rather than relying on statistical regularities specific to either visual observations or instructions. The effectiveness of our approach has been illustrated in two challenging VLN tasks. Crucially, our method is a general model that can be implemented not only in any VLN task but also in complex multi-modal problems where high-level reasoning is required and generalisation is paramount; thus, we consider exploring this avenue further in future.

**Acknowledgements**

This work was partly supported by Australian Research Council grant DP160100703. This material is based on research sponsored by Air Force Research Laboratory and DARPA under agreement number FA8750-19-2-0501. The U.S. Government is authorised to reproduce and distribute reprints for Governmental purposes notwithstanding any copyright notation thereon.

**Broader Impact**

Vision-and-language navigation is a significant step in realising practical robots that can interact and follow instructions. These robots have applications in a wide range of problems including but not limited to (1) the need for tools that can operate in risky environments that human presence is dangerous is more than ever (e.g. with the recent pandemic in the health centres); (2) assistant to individuals in need, e.g. blind and disabled; (3) agriculture and manufacturing where the labour-intensive jobs require instruction following robots; etc.

Beyond the application of this paper to VLN, better generalisation in machine learning using a small training set is desired for improved performance and usability. This requires machine learning approaches that can anticipate what they might encounter when deployed. We believe counterfactuals provide a means for better utilisation of the training data, improved generalisation and even explainability. Counterfactuals, as were used in the paper, can provide more robust models that are safer to deploy since the sources of spurious bias are reduced. Moreover, these models are less prone to be affected by the bias (e.g. social) in the human-generated training data. This paper provides an early step in this direction by formalising the problem in a practical setting.

## Footnotes

[1]We model $\pi_{\boldsymbol{\theta}}$ as a recurrent model. For the language command, we use a separate recurrent model.

[2]We consider the hidden state of the agent's policy as $\mathbf{s}_t$.

[3]Our evaluation on the test set is available at: `https://evalai.cloudcv.org/web/challenges/challenge-page/97/leaderboard/270`

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
