[Supplementary Material]

# Counterfactual Vision and Language Navigation: Unravelling the Unseen

**Amin Parvaneh, Ehsan Abbasnejad, Damien Teney,**
**Javen Qinfeng Shi, Anton van den Hengel**

Australian Institute for Machine Learning
University of Adelaide, Australia
{amin.parvaneh, ehsan.abbasnejad, damien.teney,
javen.shi, anton.vandenhengel}@adelaide.edu.au

## 1 Background on Counterfactuals

We provide a brief background on counterfactuals. Further details can be found in [1].

**Definition 1** (Structural Causal Model (SCM)). A structural causal model $\mathcal{M}$ over variables $\mathbf{X} = \{X_1, \ldots, X_n\}$ consists of a set of independent (exogenous) random variables $\mathbf{U} = \{\mathbf{u}_1, \ldots, \mathbf{u}_n\}$ with prior distributions $P(\mathbf{u}_i)$ and a set of functions $f_1, \ldots, f_n$ such that $X_i = f_i(\mathbf{PA}_i, \mathbf{u}_i)$, where $\mathbf{PA}_i \subset \mathbf{X}$ are parents of $X_i$. Therefore, the distribution of the SCM, which is denoted $P^{\mathcal{M}}$, is determined by the functions and the prior distributions of exogenous variables.

Inferring the exogenous random variables based on the observations, we can intervene in the observations and inspect the consequences.

**Definition 2** (Interventions in SCM). An intervention $I = \text{do}\big(X_i := \tilde{f}_i(\tilde{\mathbf{PA}}_i, \mathbf{u}_i)\big)$ is defined as replacing some functions $f_i(\mathbf{PA}_i, \mathbf{u}_i)$ with $\tilde{f}_i(\tilde{\mathbf{PA}}_i, \mathbf{u}_i)$. The intervened SCM is indicated as $\mathcal{M}^I$, and, consequently, its distribution is denoted $P^{\mathcal{M};I}$.

The counterfactual inference with which we can answer the "what if" questions will be obtained in the following process:

1. Infer the posterior distribution of exogenous variable $P(\mathbf{U}_i | \mathbf{X} = \mathbf{x})$, where $\mathbf{x}$ is a set of observations.

2. Replace the prior distribution $P(\mathbf{u}_i)$ with the posterior distribution $P(\mathbf{u}_i | \mathbf{X} = \mathbf{x})$ in the SCM. We denote the resulted SCM as $\mathcal{M}_\mathbf{x}$ and its distribution as $P^{\mathcal{M}_\mathbf{x}}$

3. Perform an intervention $I$ on $\mathcal{M}_\mathbf{x}$ to reach $P^{\mathcal{M}_\mathbf{x};I}$.

4. Return the output of $P^{\mathcal{M}_\mathbf{x};I}$ as the counterfactual inference.

(a) VLN SCM        (b) VLN SCM with interventions

Figure 1: Structural Causal Model (SCM) of the vision-and-language navigation (VLN). We incorporate an exogenous variable in the SCM that is learned and utilised for reasoning about interventions in the observation.

## 1.1 Counterfactual Vision and Language Navigation

We concentrate on the interventions on the visual observations to improve the generalisation of the model to the unseen environments. Our intuition is that the visual feature extractor functions in VLN usually focus on spurious features in the scene. To that end, by constructing the SCM for VLN and introducing interventions in the training environments, we train models that better generalise to unseen environments.

Fig. 1 shows the SCM for VLN at a time-step. The SCM consists of an exogenous variable $\mathbf{u}$ (for observations) and a set of functions that transmit the observation $\mathbf{o}_t$, language instruction $\mathbf{c}$ and previous state $\mathbf{s}_{t-1}$ to the next state $\mathbf{s}_t$ and, subsequently, to the next action $a_t$. We intervene in observations by replacing their embedding function $f_{\mathbf{o}}$ with $\tilde{f}_{\mathbf{o}}$. Specifically, after learning exogenous variable $\mathbf{u}$, we replace the latent representation of the observations $\mathbf{z}_t$, to $\tilde{\mathbf{z}}_t^{\mathbf{u}}$.

In Eq. (5) in the paper, we effectively remove the effect of the intervention leading to an agent that is less biased towards spurious features. For computing the expectation, we could take samples from the posterior $p(\mathbf{u} \mid \tau, \mathbf{c})$ and average using multiple counterfactual trajectories (an MCMC approach). Instead, in the interest of efficiency in Sec 3.3, we take only one instance from the mode of the posterior that alters the navigation policy's output.

# 2 Implementation Details

## 2.1 R2R Navigation

Our navigation policy is a attention encoder-decoder network that encodes the navigation history conditioned on the instruction and decodes the next direction that the agent should follow. To have a fair comparison and show the effectiveness of our approach, we closely follow the implementation proposed by [2] and evolved in [3]. Our encoder is a recurrent neural network:

$$\mathbf{h}_i^e = f^e(f^w(w_i), \mathbf{h}_{i-1}^e), \tag{1}$$

where $f^w$ represents an embedding layer, $f^e$ is a bidirectional LSTM and $\mathbf{h}_i^e$ is the latent representation vector for word $i$ in the instruction ($\mathbf{h}_i^e \in \mathbb{R}^{512}$), which is obtained from the concatenation of forward and backward layers of the LSTM.

We calculate the attention over a collection of $V$ values ($\mathbf{v}_i$) with respect to a key vector ($\mathbf{k}$) as:

$$\alpha, \mathbf{att} = \text{Attention}(\mathbf{k}, \{\mathbf{v}_i\}_{i=1}^V), \tag{2}$$

$$\text{with} \quad \alpha_i = \text{Softmax}(\mathbf{v}_i^\intercal \mathbf{W} \mathbf{k}_i), \quad \mathbf{att} = \sum_{i=1}^V \alpha_i \mathbf{v}_i,$$

where $W$ are the parameters to be learned, $\alpha_i$ is the weight of $i$-th value item and $\mathbf{att}$ is the attentive feature vector.

Our decoder is an attentive RNN:

$$\_, \hat{\mathbf{z}}_t = \text{Attention}^v(\mathbf{h}_{t-1}^d, \{\mathbf{z}_i^t\}_{i=1}^{36}), \tag{3}$$

$$\mathbf{h}_t^d = f^d([f^a(a_{t-1}); \hat{\mathbf{z}}_t], \mathbf{h}_{t-1}^d), \tag{4}$$

$$\_, \hat{\mathbf{h}}_t^d = \text{Attention}^l(\mathbf{h}_t^d, \{\mathbf{h}_i^e\}_{i=1}^L), \tag{5}$$

$$\{p_j\}_{j=1}^N, \_ = \text{Attention}^d(\hat{\mathbf{h}}_t^d, \{\mathbf{z}_j^t\}_{j=1}^N), \tag{6}$$

where $\mathbf{z}_i^t$ is the concatenation of 2048-dimensional visual feature vector (extracted from a pretrained ResNet[4]) and a 128-dimensional angle embedding vector, $f^a$ is an embedding layer to embed the previous action into a 64-dimensional vector, $f^d$ is another LSTM, and $\hat{\mathbf{h}}_t^d \in \mathbb{R}^{512}$ represents the language-grounded state of the navigation. The action is chosen greedily or by sampling (in IL or RL setting respectively) among the $N$ possible movable directions based on their corresponding weight $p_j$. It worth mentioning that we apply a Dropout of $0.5$ between all layers of the network.

## 2.2 EQA Navigation

To attend the visual features of the egocentric RGB image in House3D environments, we utilise the pre-trained CNN proposed in [5]. The network consist of 4 convolutional blocks in which a $5 \times 5$

convolution layer is followed by BatchNorm, ReLU and $2 \times 2$ MaxPool layers. The network is trained in a multi-task learning setting where the outputs of the last convolutional block are fed into three separate decoder heads for RGB image reconstruction, pixel-wise semantic segmentation and semantic classification. In our experiments, we extract the outputs of the last convolutional block ($\mathbb{R}^{3200}$) and downsize its dimension to 128 using a fully-connected layer to reach latent observation representations $\mathbf{z}$.

Our navigation policy is a recurrent encoder-decoder model. We encode the question using a 2-layer LSTM. The last hidden state of the encoder is used as the embedding of the whole question ($\mathbf{h}^e \in \mathbb{R}^{64}$). The decoder is an RNN followed by MLP and LogSoftmax layers:

$$\mathbf{h}_t^d = f^d\big([f^a(a_{t-1}); \mathbf{h}^e; \mathbf{z}_t], \mathbf{h}_{t-1}^d\big), \tag{7}$$

$$\mathbf{p}_t = \text{LogSoftmax}\big(f'^d(\mathbf{h}_t^d)\big), \tag{8}$$

where $f^a$ is the action embedding of dimension 32, $f^d$ is a 2-layer GRU with a hidden state of size $1,024$, $f'^d$ is a fully-connected layer mapping the outputs to the action space, and $\mathbf{p}_t$ ($\mathbf{p}_t \in \mathbb{R}^4$) declares the probability of each action at step $t$.

## 2.3 Counterfactual Learning

The learning process of exogenous variables $u$ for two samples picked from the dataset ($\{(\tau, \mathbf{c}), (\tau', \mathbf{c}')\} \sim \mathcal{D}$) is as follows:

1. Repeat the last observation of $\tau'$ to be the same length as $\tau$.
2. Sample $\mathbf{u}$ from the prior distribution Beta$(0.75, 0.75)$.
3. Generate counterfactual visual features using $\mathbf{u}$ and based on Eq. 9 (in the main text).
4. Feed the counterfactual trajectory into Speaker and Navigator.
5. Update $\mathbf{u}$ based on Eq. 11 (in the main text) with learning rate of 0.1.
6. Repeat steps 3 to 5 for $N$ iterations ($N = 5$ in the experiments).

## 2.4 Reinforcement Learning Reward

The reward function we use is measured based on both agent's progress toward the target location and its final success/failure. To that end, at each step we calculate the distance to the target location ($d_t$) and, based on that, we measure the progress reward ($d_t - d_{t-1}$). Additionally, at the end of each episode (either by reaching the maximum number of steps or after choosing the `stop` action), if $d_t$ is lower than 3 meters, we provide the agent with a big reward of size $+2$. Otherwise, we punish the agent with a negative signal of $-2$. Note that we set the discount factor to 0.9 in all experiments.

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

**Instruction:** *Walk forward up the set of three stairs. Enter the room at the end of the hallway. Walk o the massage table, and stop.*

Figure 2: Trajectory Sample. The baseline agent (left trajectory) follows the language instruction until step 4, where instead of moving towards the *massage table*, it goes into the next *hallway*. We argue that since there are limited *massage tables* in the training set, the baseline method does not consider this one as a variant of table and continues searching to stop at the end of the next hallway (6 meters away from the target position). On the other hand, our agent (at the right side), succeeds in identifying the *massage table* and ends up at the target location without any navigation error.

**Instruction:** *Walk through double doors into the house. Continue around the dining table and through the entry way to the next room. Walk up to the couch and armchairs surrounding a coffee table.*

Figure 3: Trajectory Sample. Both agents follow the same path until step 3, where they need to identify and reach the *coffee table* that is surrounded by the *couch* and *armchairs*. In contrast to the baseline model that looks for typical tables in the environment and overlooks the couch from the back view, our model, recognises them and attends the target position successfully.

**Instruction:** *'Exit the room then go straight and turn left. Go straight until you pass an eye chart picture frame on the left wall then wait there.*

Figure 4: Trajectory Sample. The baseline model neglects a part of the instruction, and it seems that it has presumed the *picture frame* in step 2 as the one mentioned in the guidance improperly. Biased by the great number of trajectories in the training set, it decides to go into the door at the opposite side, which costs the agent to end up unsuccessfully (7 meters away from the target). On the contrary, our approach executes the instruction precisely, finds the *eye chart picture frame* correctly, and stops at the vicinity of the goal location (1 meter error).

**Instruction:** *Walk straight past the bar through the doorway. Turn right at the picture and enter the bedroom. Stop and wait by the closet.*

Figure 5: Trajectory Sample. From the pictures it is evident that the baseline approach cannot find the right path, that is identifiable with the *picture* clue in the instruction, and, consequently, ends up about 13 meters away from the target. On the other hand, our approach succeeds in correlating the language instruction to the correct path and reaching the target location (less than 1 meter error).