[Reviews · NeurIPS 2020]

Review 1

Summary and Contributions: Paper presents an approach to generate counterfactual observations along a trajectory (in the context of VLN and EQA tasks, but the approach appears to be generally applicable). The core of the idea relies on mixing observations (in a learned proportion) along two different randomly sampled trajectories such that it encourages different actions.

Strengths: - Approach is well-motivated and sensible. - Paper is well-written. - Experimental evaluation is reasonably thorough (see some minor comments below). Results are generally positive. Not ground-breaking, but consistently positive.

Weaknesses: - No statistical significance (of results) is reported. - Approach shares high-level similarity with Mixup [35,36] and with [34]. Wrt Mixup, I am perplexed by the explanation in L95-97: why is picking a mixing parameter in Mixup any more difficult than picking alpha in the proposed approach? Also, what would a direct application of Mixup to this task look like? Would it just involve sampling 2 random trajectories and using a dataset-level mixing parameter? That would be worth comparing to, to establish the efficacy of the per-sample learned mixing approach being proposed here. Wrt [34], could the authors explain what an translation of equations 2 and 3 from [34] would look like for VLN? Is it essentially the approach being proposed here? If not, what would the differences be? And is it possible to compare to it? ---------------------- Post Rebuttal ----------------------- Thank you for your response. I find it problematic that the author response does not address fairly specific questions about unsubstantiated assertions in the manuscript and problematically adds more such assertions in the author response. Specifically: > Wrt Mixup, I am perplexed by the explanation in L95-97: why is picking a mixing parameter in Mixup any more difficult than picking alpha in the proposed approach? This question was not answered in the response. > (2) an interpolation of state-action from one trajectory to another may lead to catastrophic difference in the objective. And what empirical evidence or reasoning supports this claim? Any "may" lead to a catastrophic difference. But where's the evidence for that? These points are not sufficient to prevent the publication of this manuscript, but I strongly encourage the authors to remove such unsubstantiated claims from the final version.

Correctness: Empirical claims appear well supported. Mathematical claims are few but appear correct.

Clarity: Generally okay. Please consider not using variables before describing them (e.g. but not limited to z in (1)). N in Alg 1 is not specified.

Relation to Prior Work: See comments above.

Reproducibility: No

Additional Feedback: - Please fix inconsistent and inaccurate boldings in Table 1 and Table 2. E.g. in Table 1 IL+RL+Aug+Counterfactuals is bolded for NL, when IL+RL+Aug* is lower/better. There are multiple other such inconsistencies/inaccuracies. - Please be aware that SUNCG (used in 4.2) is currently unavailable pending a lawsuit from Planner5D. This renders the experiments in 4.2 essentially non-reproducible. Please consider switching to the EQA dataset on Matterport scenes introduced here: https://arxiv.org/abs/1904.03461. Of course, this factor should not play any role in making a decision on this submission.


Review 2

Summary and Contributions: This paper introduces a method for generating *counterfactual* visual features for augmenting the training of vision-and-language navigation (VLN) models (which predict a sequence of actions to carry out a natural language instruction, conditioning on a sequence of visual inputs). Counterfactual training examples are produced by perturbing the visual features in an original training example with a linear combination of visual features from a similar training example. Weights (exogenous variables) in the linear combination are optimized to jointly minimize the edit to the original features and maximize the probability that a separate speaker (instruction generation) model assigns to the true instruction conditioned on the resulting counterfactual features, subject to the constraint that the counterfactual features change the interpretation model's predicted timestep at every action. Once these counterfactual features are produced, the model is trained to encourage it to assign equal probability to actions in the original example when conditioning on the original and the counterfactual features (in imitation learning), or to obtain equal reward (in reinforcement learning). The method improves performance on unseen environments for the R2R benchmark for VLN, and also shows improvements on embodied question answering. --- update after author response --- Thanks to the authors for the response. After considering the response, the other reviews, and some discussion, I feel more positively about the paper and have raised my score. Re: the results, it's encouraging that the parameters for the prior were tuned. I am convinced by the claim in the response that the improvements of the full method over this prior are likely to be real and substantial, given the saturation on the dataset, but I also agree with R1 that a significance test would be helpful here to confirm. I agree with the other reviewers that the lack of clarity of the presentation (R3) and the lack of qualitative analysis (R4) are still flaws, but given the clarifications in the response I'm optimistic that the clarity could be addressed in any future version of the paper.

Strengths: Applying data augmentation to visual features to improve generalization to unseen visual contexts is well-motivated, given a range of prior work on e.g. VLN showing that models overfit to features of seen environments. This will likely be of interest to researchers in this area. I found the method interesting, as a non-trivial and novel way to construct an adaptive version of the environmental dropout of Tan et al., and to extend the counterfactual learning method of Abbasnejad et al. to sequential inputs and outputs. The method shows consistent improvements to a state-of-the-art baseline model on unseen environments and across training conditions in VLN, and also shows improvements on EQA.

Weaknesses: It seems that a natural baseline to compare against would be to inject random noise into the observations to produce the counterfactual observations. The experiments with +Prior capture this, but would the results improve with a different choice of prior? i.e. was grid search performed to tune the parameters of the Beta distribution for the Prior experiments, and then the counterfactual experiments applied on top of this? The improvement of the counterfactual method over the Prior method (which seems much simpler to implement, as it doesn't need the speaker model or the inner loop optimization of u) is relatively small (around 1 point SR and 1-2 points SPL in seen environments for VLN), so that it's unclear to me whether (in its current form) the full method would be adopted (over just noising the features).

Correctness: The formulation of the method up through Eq 8 in section 3.2 seemed correct to me. Section 3.3 (the description and derivation of Eq 11) was unclear to the extent that I'm not unsure about the correctness -- see clarity below. However, the approximation that is actually used (Eq 11) seems reasonable. The evaluation seemed correct and in line with standards for R2R, and the base models used seemed reasonable. Lines 268 and 271 claim significant improvements; but are these statistically significant?

Clarity: I found the description in 3.3 difficult to follow. See below for some specific important questions and points, and "additional comments" for some minor points: The sampling of the trajectory pairs (170: "We first sample a pair of real trajectories from the observations such that at least one has the same instructions as given") was unclear. I'm unsure how to interpret this other than that two pairs are sampled with (\tau, c) and (\tau', c) [i.e. c is shared across both pairs], but this doesn't match the notation -- and given the freeform annotation process for R2R (AMT workers wrote long instructions, typically multiple sentences, for each trajectory), it seems unlikely that a given instruction in the dataset was produced as an annotation for multiple different trajectories. The interpretation of the u variables, and description of the posterior over u and how it is used, were also unclear. I'm not sure how well-motivated the probabilistic presentation of u is, given the approximations and assumptions used. - 167: "Here, u is marginalised out", but from the description in 3, it seems that only an approximation to marginalization (optimizing toward u close to 1, as a proxy for having a high posterior probability) is used. - 186: "We seek the minimum intervention i.e. minimum edit": is this assuming that high posterior u's correspond to minimum edits? Or assuming that u's close to 1 correspond to minimum edits (which seems more supportable)? - 194: "The first term is maximized when u is close to one": does this refer to the posterior p(u | \tau, c)? I can see how this is true for the likelihood ~\pi_\theta(\tau | c, u), but it's unclear to me why this is also true for the posterior unless the particular parameters used for the Beta prior produce this. The high-level description of the method was a bit scattered between the abstract, intro, Figure 1 caption, and related work. The summary section above gives my understanding, but I'm not sure it's correct. I think it would be helpful to, in one place, say how the counterfactual examples are generated and how the actions are supervised for each counterfactual example.

Relation to Prior Work: The description of the relation to prior work was reasonable and informative. I wasn't clear on whether the base model results and this model when using the prior are comparable, given that IL are taken from prior work (Tan et al.), and the "Baselines" section only says that the setting is similar to [11].

Reproducibility: No

Additional Feedback: The description of the states s_t in section 3.1 was confusing. I thought they might be produced by a recurrent decoder ("s_t is a representation of the trajectory history", and its use throughout the rest of the paper), but the notation in Eq (1) is confusing since p(s_t | ...) is not written as a function of \theta, and states in a POMDP (line 104) are typically the underlying hidden states (which the observations are functions of), not functions of the observations. I also had trouble squaring this with line 110, "s_0 is a sample from the state prior". line 142: "The exogenous variable is conditioned on the factual trajectories", but from Fig 2 in the supplementary, "u" is an ancestor of all nodes in the SCM. Does this mean conditioned through posterior inference? 189: "we additionally constrain the counterfactual trajectory to have the same instructions": are these the same constraint mentioned in 170, or something different? The claim in the conclusion that the method forces the model to learn semantics and high-level relations rather than relying on statistical regularities wasn't fully supported -- I suppose the decreased generalization gap gives some weak evidence for this, but I don't think it's possible to say that this a better representation of semantics or high-level relations is why the models are generalizing better. Some additional points that should be clarified given more space: - 159: "allows the model to focus on correct actions at each state rather than variations": this is unclear to me, it would help to expand more on this. - Is the speaker model p(c | ~\tau) is fixed or updated during training? - Do the IL experiments use student forcing or teacher forcing? Minor points: - Eq (6) and line 155 use \epsilon_\tau twice (instead of \epsilon_d?) - 164: "hinders on" -> "hinges on" - 259: Pior -> Prior - 320-329: minor grammar errors - Should BC+RL* in Table 1 be IL+RL? If not, are these results comparable to IL+RL+{Prior,Counterfactuals}?


Review 3

Summary and Contributions: Post Rebuttal: I thank the authors for addressing my concerns and after discussions with other reviewers, I have raised my score. ---------------------- This paper proposed a method for training VLN navigation agents based on utilizing an additional loss that encourages the agent to better handle a counterfactual trajectory

Strengths: - The authors propose a novel method for improving the generalization of VLN agents by optimizing a counterfactual trajectory in addition to the true trajectory - The proposed method outperforms existing VLN methods trained with similar data and is very competitive with PREVALENT, which uses external data for pre-training - The proposed method also outperforms existing approaches on EQA - Overall, the proposed method is intuitive (at a high level) and works well

Weaknesses: - The proposed method is computational expensive. - The paper is not clearly written, in particular section 3.3 (I will expand more upon this in the Clarity section)

Correctness: Yes

Clarity: - Section 3.3, in particular the paragraph about sampling from pi(..) is confusing, particularly for RL. I think is that U is chosen given (tau, c) and (tau', c') and then z is generated using tau' AND the actually trajectory taken from the policy? - In the case of teacher-forcing, I don't understand "such that at least one has the same instructions as given". What does "as given" refer to?

Relation to Prior Work: Well situated with respect to prior work

Reproducibility: No

Additional Feedback: Overall, this is an interesting paper but the clarity in writing needs to be improved before publication


Review 4

Summary and Contributions: This paper presents a new method for generating counterfactual data in VLN and EQA. It introduces an exogenous variable that controls changes applied to the input visual representations. This variable is chosen so that the agent's action is altered but the resultant counterfactual trajectory is still a valid execution of the language instruction. The agent is trained to maximize performance on both the original and the counterfactual trajectories. Results on both VLN and EQA shows improvement upon baselines.

Strengths: The problem studied is highly important and practical. This presented framework is general and can potentially be applied to other sequential decision-making tasks. The paper provides a rigorous derivation of the proposed framework. Experiments thoroughly compare the proposed framework with pre-existing ones. Results strongly support the effectiveness of the framework.

Weaknesses: First, this framework is presumably computationally expensive. The reviewer would like to see a discussion of the computational cost of training each model (ideally, a new column in the result tables). It may be also good to conduct an ablation study on the effect of N (the number of gradient updates on u) on the results. Second, the paper is missing quantitative/qualitative analyses of the counterfactual data and their effects on the agent's decisions. The reviewer would like to see evidence that (a) the counterfactual data actually alter the agent's decisions and (b) learning with counterfactual data helps the agent generalize better in specific scenarios. Otherwise, the reviewer is not convinced that the framework improves performance of the agent for the advertised reasons.

Correctness: ** Notation confusions ** First, what exactly is the state s? Is it a state of an RNN or the VLN simulator? If the former is the case, these states should not be a part of the dataset (in line 115, the dataset provides tau_i, which contains states s_t). In both cases, how is the transition distribution p(s_t, | s_{t - 1}, z_t, c) defined? If s is an RNN state, where is the stochasticity in the RNN? If s is a simulator state, how can p incorporate both a continuous vector z_t and a simulator state s_{t - 1} to generate a new simulator state s_t? How can this p be trained via back-propagation (as equation 11)? Second, the action a in equation 11 is not defined anywhere in the equation. Did you mean a_t? Third, in algorithm 1, which tau~ is used to compute the imitation learning gain among N tau~ that are generated during the for loop for computing u? Please also add line numbers to this algorithm box. Finally, equation 4 says that u is sampled from p(u | tau, c) but it seems like u is the result of an optimization problem rather. ** Results ** The SR and SPL results for PRESS and PREVALENT in table 2 are off the scale. Please correct them.

Clarity: The paper is well-written and easy to follow. There are a few notation confusions that need to be addressed (see correctness).

Relation to Prior Work: Yes. The paper clearly discusses the differences compared to a recent work on counterfactual vision language [1]. [1] Ehsan Abbasnejad, Damien Teney, Amin Parvaneh, Javen Shi, and Anton van den Hengel. Counterfactual vision and language learning. In The IEEE Conference on Computer Vision and Pattern Recognition (CVPR), June 2020.

Reproducibility: No

Additional Feedback: ** After author response **: Many thanks for the response. I have read the qualitative examples in the supplementary materials. However, I am still not satisfied by the explanations of the examples. It is clear that the proposed method is better at detecting rare objects, but I still had a hard time understanding how being to counter-factually reason helps improve that capability.

[Author Response · NeurIPS 2020]

We would like to thank all the reviewers for their insightful comments. All reviewers confirm the contribution of our
paper in providing a novel framework for improving the generalisation of VLN tasks in addition to a sound formulation
with wider range of applications. Please see below for specific responses.

**Response to Reviewer #1:**

**Relation to Mixup:** Mixup is not directly applicable to VLN since (1) it is sequential in nature, (2) an interpolation of
state-action from one trajectory to another may lead to catastrophic difference in the objective. Our approach intervenes
in the visual features to simulate the agent's behaviour in a counterfactual environment, where the agent still has to
follow the same instruction and sequence of actions.

**Difference with [34]:** It is the closest approach to ours and general differences are highlighted in L89-93. Specifically
for counterfactual distribution learning, similar to theirs, our approach creates counterfactual samples that are hard for
the agent to follow with minimum interventions. However our approach is different to [34] in (1) ours is formulated
to minimise the expected difference of the model on observations and the intervened samples while [34] resorted to
importance sampling that could have a large variance (in fact it was not as successful in VLN); (2) ours is sequential;
(3) our instructions have actions as opposed to vision and language only (4) we had a model of a speaker to incorporate.

**Response to Reviewer #2:**

**Prior distribution and random noise:** The hyper-parameters of the prior distribution have been selected through a
simple grid search before applying the counterfactual distribution learning. Additionally, Environment Dropout [11]
can be considered as a random noise injection which has been reported in Table 1 and 2.

**Improvements:** Generally, the reported improvements are indeed significant (around $4\%$ in success rate and SPL),
considering the fact that R2R task has been explored extensively in recent years so that even large-scale self-supervised
pre-training [44] gains less than $4\%$ improvement. By adding an insightful though simple counterfactual learning
process, we gain further $1-2\%$ improvement on top of the improvement gained using the prior ($2-3\%$), which is a
significant improvement on top of a model that enjoys a better generalisation than other baselines.

**Counterfactual Learning:** For the counterfactual distribution learning, we sample two pairs of real trajectories and
only use language instruction of the first one. We will clarify this in the camera-ready version. In L167, we are
approximating the marginalisation of $\mathbf{u}$ by adjusting the variable generated from the prior, instead of relying on costly
methods like MCMC or a variational lower bound. It is also worth mentioning that as stated in L180 and according
to Eq. 9, minimum edit (intervention) happens when $\mathbf{u}$ is close to one. In addition, for L194, please note that as
stated in L178, $p(\mathbf{u} \,|\, \tau, \mathbf{c}) \propto p(\mathbf{u})\tilde{\pi}_{\boldsymbol{\theta}}(\tau \,|\, \mathbf{c}, \mathbf{u})$. Therefore, $p(\mathbf{u} \,|\, \tau, \mathbf{c})$ is maximised when $\mathbf{u}$ is close to one. Rather than
integrating $\mathbf{u}$ out, we choose the most likely sample corresponding to a counterfactual. It should be noted that Figure
2.a in supplements is revealing the general SCM in VLN. Starting from a prior distribution, this variable is learnt
conditioned on the inventions on the real observation and then is used for the counterfactual generation (Figure. 2.b).

**State variable**: In our experiments, $\mathbf{s}_t$ is the hidden state of the RNN model and the parameters are omitted for brevity.

**Training setting:** We freeze the speaker model during the counterfactual distribution learning. Additionally, as stated
in L231, following the same setting as [11] and [8] for a fair comparison, we use teacher-forcing during IL.

**Response to Reviewer #3:**

**Computational cost:** Thanks for mentioning an interesting point. The computational cost of our proposed method
highly depends on the number of iterations for learning the counterfactual distribution. Higher values result in better
approximations of the exogenous variable while increasing the training time. We undertook several experiments for
finding a point of best trade-off and found that even few iterations (5 as reported in Section 3.1 in the supplements)
could contribute to a good understanding of the variable and therefore better results.

**Sampling for RL:** As indicated in Algorithm. 1, after finding the optimum value of $\mathbf{u}$ using $\{\tau, \mathbf{c}, \tau'\}$ and Eq. 11, at
each step of the policy rollout, we intervene the observation using the learned $\mathbf{u}$ and $\tau'$.

**Response to Reviewer #4:**

**Computational cost:** Please refer to the response to reviewer #3.

**The effects of counterfactual data:** We have demonstrated some trajectories revealing the difference between the
baseline and our model over the same starting point and the same instruction in supplements. In addition, using our
approach we observe a significant improvement in the test set that highlights the positive effect of our approach.

**State variable:** Please refer the response to reviewer #2. We will clarify in the camera-ready.

**Equation 11:** Thanks for pointing out the typo. We will replace $a$ with $a_t$ in the camera-ready version.

**Algorithm 1:** In fact, we use the optimised $\mathbf{u}$ to generate a new $\tau$ based on Eq. 9 for IL gain calculation. We haven't
repeated the counterfactual generation for brevity of the algorithm.

**Equation 4**: As is mentioned in L182-185, for simplicity and efficiency, $\mathbf{u}$ is marginalised out from the Eq. 4 and is
approximated using Eq. 11.

[Meta-Review · NeurIPS 2020]

After reading each other's reviews and discussing the author response, all reviewers advocate for acceptance of this submission and I agree. The proposed approach is a novel means of augmenting VLN training data and yields strong results.